# Quantifying the association between psychological distress and low back pain in urban Europe: a secondary analysis of a large cross-sectional study

Chukwuebuka Immanuel Ugwu ,[1] Daniel Pope[2]

[1]Department of Public Health and Policy, Institute of Population Health Sciences, University of Liverpool, Liverpool, UK
[2]Department of Public Health, Institute of Population Health, University of Liverpool, Liverpool, UK

**Correspondence to**
Dr Chukwuebuka Immanuel Ugwu; mazibukaz@gmail.com

## ABSTRACT

**Objectives** This study aims to estimate the prevalence of low back pain (LBP) in Europe and to quantify its associated mental and physical health burdens among adults in European urban areas.

**Design** This research is a secondary analysis of data from a large multicountry population survey.

**Setting** The population survey on which this analysis is based was conducted in 32 European urban areas across 11 countries.

**Participants** The dataset for this study was collected during the European Urban Health Indicators System 2 survey. There were a total of 19 441 adult respondents but data from 18 028, 50.2% female (9 050) and 49.8% male (8 978), were included in these analyses.

**Primary and secondary outcome measures** Being a survey, data on the exposure (LBP) and outcomes were collected simultaneously. The primary outcomes for this study are psychological distress and poor physical health.

**Results** The overall European prevalence of LBP was 44.6% (43.9–45.3) widely ranging from 33.4% in Norway to 67.7% in Lithuania. After accounting for sex, age, socioeconomic status and formal education, adults in urban Europe suffering LBP had higher odds of psychological distress aOR 1.44 (1.32–1.58) and poor self-rated health aOR 3.54 (3.31–3.80). These associations varied widely between participating countries and cities.

**Conclusion** Prevalence of LBP, and its associations with poor physical and mental health, varies across European urban areas.

## INTRODUCTION

In the past few decades, low back pain (LBP) has become increasingly recognised as a huge public health concern whose impact is expected to grow as life expectancy continues to improve around the world. At present, it is a leading cause of disability[1] and ranks among the top 10 causes of overall disease burden globally.[2] LBP represents a substantial burden to both the society and the individual. The costs to society include associated utilisation of primary care and other health services for prolonged treatments, complex physical and surgical interventions, resulting in enormous

### STRENGTHS AND LIMITATIONS OF THIS STUDY

⇒ A key strength of this study is that it is based on uniformly collected data from a large population survey conducted in 32 European cities.
⇒ Given that exposure and outcome data were collected simultaneously, our study is unable to establish a temporal sequence.
⇒ Our analysis does not explore a dose–response relationship between low back pain and psychological distress.
⇒ All variables included in this analysis are comparable across the different settings.

financial costs to health services.[3 4] LBP is also a major reason for temporary or permanent shortages in the workforce through sickness absence with sufferers requiring disability benefits, having a direct negative effect on the economy.[5]

In terms of burden to individuals, there is an increased need for supportive care and reduced incomes following worklessness, LBP can thus constitute a strain on family resources.[6] Also, it significantly interferes with a person's ability to engage in meaningful active social life predisposing sufferers to further morbidity and reducing individuals' overall quality of life.[7] Despite increasing treatment options such as pharmaceutical options, physical therapy, spinal manipulation and surgery,[8] improvements in outcomes may be modest at best.[9]

Psychological distress is a known mental health problem quite common in modern populations. It has been shown to play a key role in both the aetiologic and prognostic courses of LBP in adults.[10–12] Psychological distress impacts greatly on response to LBP therapy,[10] development of disability[11] and chronicity[13] of LBP all of which have significant economic implications.[5] Conversely, background psychological distress is a recognised predictor of LBP in adults.[14]

Hitherto, however, there has not been any robust cross-national study exploring psychological distress associated with LBP. Previous attempts had limited comparability across national borders due to the heterogeneity of definitions and data collection tools, differing baseline and outcome measures as well as issues with study design,[15] limiting their usefulness. This study aims to use a large, standardised, uniformly collected dataset to estimate the burden of LBP across Europe as well as its associated mental health burden. In addition, although LBP affects all population settings, the burden is expected to differ in urban settings.[1] This is because urbanisation is associated with lifestyle changes that affect disease burden[16] and like LBP, urbanisation trend is expected to rise in the coming decades.[17] Again, no study has specifically explored psychological distress associated LBP among urban dwellers. Our key research question is: among adults dwelling in urban Europe, what is the risk of psychological distress in those with LBP compared with those without LBP.

## METHODS
### Overview of the second European Urban Health Indicator System (EURO-URHIS 2) Survey setting
This research is based on the data from the EURO-URHIS 2 conducted in 2011. The initiative was developed in an attempt to build an urban health indicator system that will help to understand the health of urban populations across Europe.[18] The project set out to develop and use standardised health survey tools and methodologies, which would be applicable to heterogeneous urban areas within the region. A key objective was to ensure the collection of reliably comparable data across urban areas on five key domains of urban health and determinants of health which include: demography, lifestyle, health status, health service utilisation and environmental health.[18]

A total of 32 urban areas across 11 countries participated in the study,[18] France (Bordeaux, Montpellier), Germany (Oberhausen), Lithuania (Kaunas, Siauliai), Macedonia (Skopje, Tetovo), Norway (Oslo), Romania (Bistrita, Craiova, Iasi), UK (Birmingham, Cardiff, Glasgow, Greater Manchester—5 urban areas, Merseyside—5 urban areas), Slovenia (Ljubljana, Bratislava, Maribor, Kosice), Turkey (Ankara, Izmir), The Netherlands (Amsterdam, Utrecht). The EURO-URHIS 2 project clearly defined comparable boundaries applicable to participating urban areas avoiding the problem of misclassifying sparsely populated urban areas and densely populated villages.[19] Standard population-based survey methods were used in the adults' research on which this work is based. Representative random samples stratified by age (18–64; ≥65) and sex (male; female) were obtained from available population registers in each urban area. Other methodological details of the EURO-URHIS 2 including the validation processes of data collection tools and the multilingual translations have been published elsewhere.[18]

### Measurement of key variables
The exposure (predictor) variable for this study was LBP while psychological distress and self-rated health were the main outcome variables. Data was also collected on potential sociodemographic confounders known to have a relationship with the predictor. These include: age, sex, level of formal education attained and socioeconomic status. All these were accounted for through multivariable regression.

The 1-month period prevalence of LBP was assessed using a prevalidated question on back pain.[20] Individuals were asked: 'In the past month, have you had LBP which lasted 1 day or longer?' with the option of indicating 'yes' or 'no'. The dependent variable psychological distress was measured using the 12-item General Health Questionnaire (GHQ).[21] The GHQ is a validated population-based screening tool for psychological distress.[22] It is considered to be the gold standard in assessing psychological distress and thus has been widely used in LBP[23 24] and non-LBP surveys.[22] Individuals were required to answer each of the 12 questions on the GHQ using a 4-point severity scale to assess psychological distress. Before analysing of the psychological distress variable, the four possible responses were dichotomised into negative (scores 1 and 2) and positive (scores 3 and 4) according to severity. So, on each item of the GHQ, a respondent will then have either 'negative' or 'positive'. Thereafter, an overall score was computed for each person with a lowest of 0 and a highest possible score of 12. Overall psychological distress score≥4 was considered significant to yield a binary outcome with scores 0–3 'no psychological distress' and scores 4–12 'psychological distress'.

The state of physical health was assessed with a 'Likert-type scale' self-rated health question in which participants were asked: 'How is your health in general' and were required to tick one of 'very good', 'good', 'fair', 'bad' or 'very bad'. Self-rated health is a known predictor of mortality.[23] The question itself is a previously validated way of estimating the general state an individual's health and has been used in population-based health surveys.[24 25] For the analysis, this variable was dichotomised and 'very good' and 'good' categories were recoded and 'good health', and the 'fair', 'bad' and 'very bad' categories were recoded into 'less than good' state of physical health. Participants who selected the 'don't know' option were recoded as missing values. Socioeconomic status was estimated using a validated generic measure that can be applicable to heterogeneous settings. Participants were required to respond 'yes' or 'no' to the question 'do you have enough money for daily expenses, for example, accommodation, travel, clothing, food?'. This approach was considered suitable as different urban areas (UAs) use dissimilar indices to express socioeconomic status or deprivation in their locales, which may not be comparable to those of other settings. The level of education was categorised as either 'no formal education or below', 'primary education', 'secondary education' or 'university education'.

## Data analysis methods

One-month period prevalence of LBP was estimated with associated 95% CI for all participating countries and cities. The prevalence was further analysed by age and sex. $\chi^2$ hypothesis tests were used to estimate statistical significance for all categorical comparisons. The prevalence of psychological distress was estimated as a dichotomised variable (a score of ≥4 representing distress—a standardised cut-off).[26] To explore the relationship between LBP and physical and mental health, unconditional logistic regression analyses were carried out. Univariable logistic regression was used to summarise the relationship between (1) LBP and self-rated health and (2) LBP and psychological distress. Multivariable logistic regression was carried out adjusting for age, sex, educational level and socioeconomic status. Comparisons of these associations between settings were illustrated using high–low–close charts. All analyses were done with SPSS V.22, Microsoft Excel and Microsoft PowerPoint V.2013.

## Patient and public involvement

Being a secondary analysis of an already existing dataset, the public and patients were not involved in the design of this study.

## RESULTS

### Distribution of LBP by place and person

The 1-month period prevalence of LBP in Europe was 44.6% (95% CI: 43.87% to 45.33%). The prevalence ranged from 33.4% (95% CI: 30.21% to 36.59%) in Norway to 67.7% (95% CI: 65.26% to 70.14%) in Lithuania (figure 1). The prevalence in Lithuania, Slovenia, Germany, Slovakia and Romania were all significantly higher than the overall European prevalence. As shown

also, the prevalence of LBP in the UK, Macedonia, The Netherlands and Norway were all significantly lower than the overall prevalence. Although the prevalence in France was lower than the overall prevalence, it did not achieve statistical significance.

Prevalence varied widely across the EURO-URHIS 2 cities (online supplemental table A). Approximately two out of three adults in eastern European cities such as Siauliai, Kaunas and Maribor reported LBP, whereas just about one in three reported symptoms Stockport, Birmingham and Skopje. Noticeably, the cities whose LBP prevalence fell below the overall European prevalence were dominated by urban areas within the UK. Of all UK cities included in the survey, only Knowsley had a prevalence rate greater than the overall prevalence.

Prevalence was significantly higher for women (48.2%) than for men (41%) (online supplemental table B). The same pattern was consistent across most of the EURO-URHIS 2 countries the only exception being Macedonia. Although male prevalence was higher in Macedonia, this was not significant. Female prevalence in Romania, Turkey, UK and the Netherlands were statistically significantly higher than those of their male counterparts. The male-to-female gap was widest in Turkey.

Generally, (as shown in online supplemental table C) the prevalence of LBP significantly increased with age. The 18–39 age group had a prevalence of 34.8%, while the 40–59 age group had 46.3%. The highest prevalence of 48.2% was recorded by over 60s. This pattern was found to be fairly consistent across all the participating countries the only exception being Germany. Furthermore, the 18–39 age group across most of the countries recorded similar prevalence rates which was around ~30%. In the UK, the Netherlands and Norway, none of the age groups

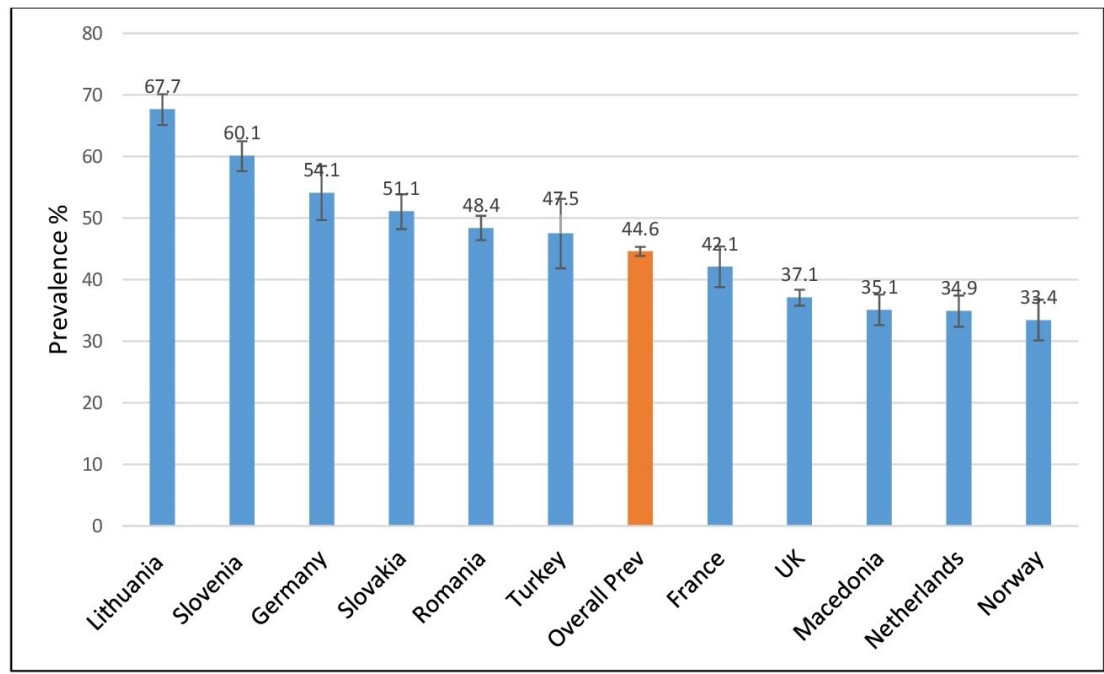

**Figure 1** Prevalence of low back pain with 95% CI in the second European Urban Health Indicator System countries.

**Table 1** Unadjusted ORs of low back pain and associated sociodemographic variables

| | N | Experienced low back pain in the last 1 month | | |
| | | % | OR | 95% CI |
|---|---|---|---|---|
| **Sex** | | | | |
| Male | 8774 | 41 | Ref | – |
| Female | 8801 | 48.2 | 1.34 | 1.26 to 1.42 |
| **Age** | | | | |
| 18–39 | 4022 | 34.8 | Ref | – |
| 40–59 | 4808 | 46.3 | 1.61 | 1.48 to 1.76 |
| ≥60 | 8745 | 48.2 | 1.74 | 1.61 to 1.88 |
| **Socioeconomic status (SES)** | | | | |
| Able to meet daily expenses | 12 569 | 40.6 | Ref | – |
| Not able to | 4064 | 57.3 | 1.96 | 1.83 to 2.11 |
| **Education** | | | | |
| University education | 5721 | 37.0 | Ref | – |
| Secondary education | 9200 | 45.9 | 2.24 | 1.85 to 2.72 |
| Primary education | 1904 | 57.2 | 2.28 | 2.05 to 2.54 |
| No formal education | 468 | 56.8 | 1.45 | 1.35 to 1.55 |

recorded prevalence rates of up to 40%. The highest differential in prevalence with respect to age was found in Romania where the prevalence doubles between the lowest and highest age categories.

On the whole, it can be seen from table 1 that people in the 40–59 age group had a 61% higher chance of LBP (95% CI: 1.48 to 1.76) than people in the 18–39 age group. Similarly, the 60 and above age group had the highest unadjusted odds for LBP, which was about 74% increased odds (95% CI: 1.61 to 1.88) compared with the youngest group. The 1-month period prevalence of LBP in the study was found to vary significantly with a generic measure of socioeconomic status and individuals' level of

education. Participants who reported not being able to meet up with their daily expenses had an almost twofold increase in risk of LBP compared with those who could (OR: 1.96, 95% CI: 1.83 to 2.11). Persons with just primary or secondary education had more than a twofold increase in the risk of LBP compared with those who had university education. Whereas those with no formal education had a 44% increase in odds (95% CI: 1.35 to 1.55).

### Associated physical and mental health burden of LBP

This study found almost a fourfold increased risk of having 'less than good' physical health in LBP sufferers compared with those who did not have LBP (shown in table 2). After accounting for the effects of age, sex, socioeconomic status and level of education, only a mild attenuation was noticed. The adjusted OR for the association was 3.54 (95% CI: 3.31 to 3.80; p value=0.0001).

This association between LBP and poor self-rated physical health was consistent across all participating countries with OR ranging from 2.14 (95% CI: 1.77 to 3.28) in France to 4.30 (95% CI: 3.19 to 5.79) in Lithuania (shown in figure 2). This association is strongest in the eastern European countries of Lithuania and Macedonia as well as in Germany.

There was also strong association between LBP and psychological distress with OR of 1.64 (95% CI: 1.50 to 1.78), indicating a 64% increase in odds of psychological distress associated with the occurrence of LBP. As shown in table 2, above, after adjusting for the effects of age, sex, socioeconomic status and level of education, the risk was slightly attenuated with a 44% elevated risk of PD among sufferers of LBP when compared with non-sufferers (95% CI: 1.32 to 1.58; p value=0.0001).

As can be seen in figure 3, the association between LBP and psychological distress varies widely between countries. It ranges from being non-existent in Macedonia and Norway to statistically significant association in the Netherlands, UK, Romania, Germany and Lithuania as indicated by their ORs and associated CIs. In the latter countries, those suffering LBP had from 39% (95% CI: 1.012 to 1.976) to 104% (95% CI: 1.4771 to 2.841) greater odds of psychological distress compared with non-sufferers. For

**Table 2** Association between low back pain and self-rated health and psychological distress

| | | 'Less than good' self-rated health | | | | |
| | | N | % | OR | 95% CI | P value |
|---|---|---|---|---|---|---|
| Low back pain | No | 3076 | 39.3 | Ref | – | – |
| | Yes (unadjusted) | 4759 | 60.7 | 3.95 | 3.71 to 4.21 | 0.0001 |
| | Yes* (adjusted) | | | 3.54 | 3.31 to 3.80 | 0.0001 |
| | | **Psychological distress** | | | | |
| Low back pain | No | 2267 | 49 | Ref | – | – |
| | Yes (unadjusted) | 2355 | 51 | 1.64 | 1.50 to 1.78 | 0.0001 |
| | Yes* (adjusted) | | | 1.44 | 1.32 to 1.58 | 0.0001 |

*Adjusted for sex, age, socioeconomic status and educational level.

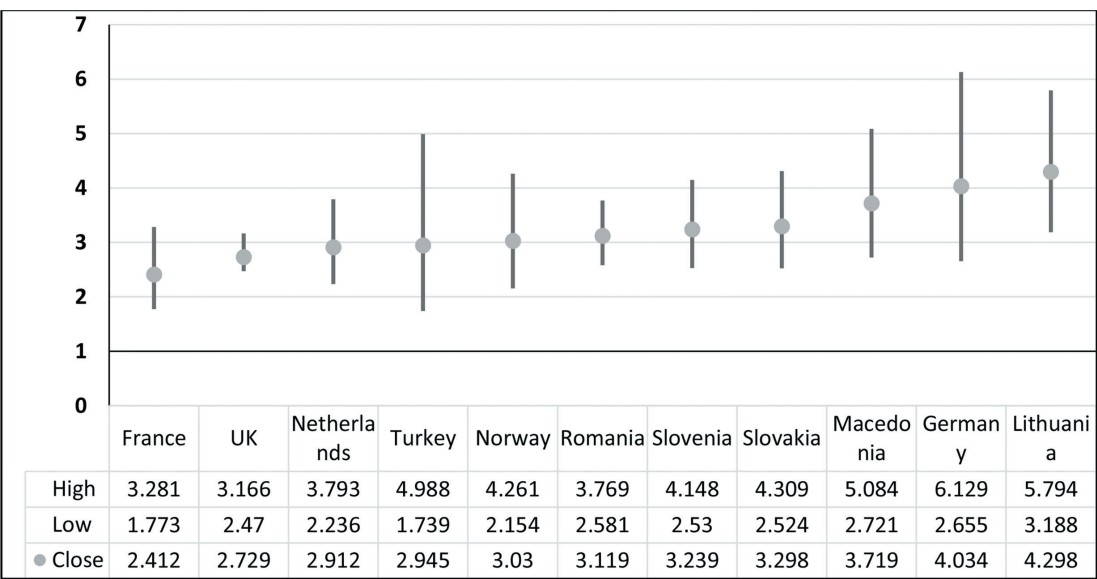

**Figure 2** Association between low back pain (LBP) and self-rated 'less than good' health. High indicates upper 95% CI. Low indicates lower 95% CI. ●Close indicates adjusted OR of the association between LBP and psychological distress (adjusted for sex, age socioeconomic status and educational level).

Turkey, France, Slovenia and Slovakia, there was a slight increase in odds of psychological distress among those suffering from LBP though not statistically significant.

Similarly, figure 4 shows that the association of LBP and psychological distress across the various urban areas varies widely from OR of 0.78 (95% CI: 0.50 to 1.21) in Skopje to 2.56 (95% CI: 1.43 to 3.57) in Siauliai.

## DISCUSSION

This research found that the 1-month period prevalence of LBP in Europe was 44.6% widely ranging between countries from 33.4% to 67.7%. This finding aligns with

findings from previous research assessing 1-month period prevalence carried out among European populations.[27–29] A Greek cross-sectional study[29] found a prevalence of 37.1% while a cohort study carried out in the north-west of England reported a 39% prevalence.[27] Furthermore, a series of five cross-sectional surveys conducted over a 20-year period in Finland with a cumulative response of 29 043 revealed an adult prevalence ranging between 46% and 51%.[28] A similar 1-month period prevalence was found by a recent systematic review.[30] However, these earlier research works were limited due to lack of comparability across heterogeneous settings. This study

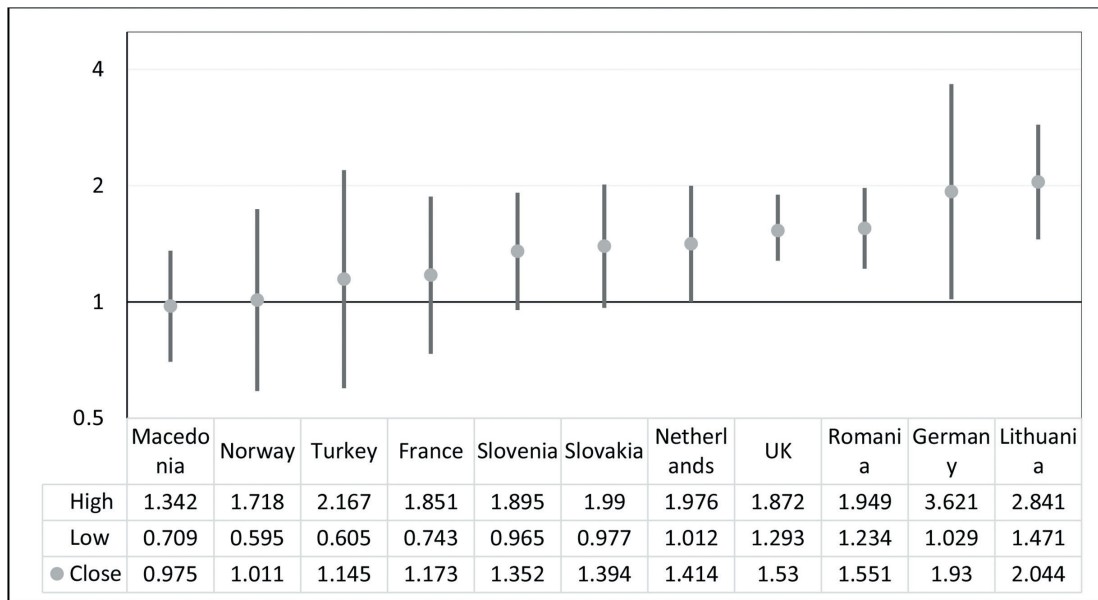

**Figure 3** Association between low back pain (LBP) and psychological distress in the second European Urban Health Indicator System countries. High indicates upper 95% CI. Low indicates lower 95% CI. ●Close indicates adjusted OR of the association between LBP and psychological distress (adjusted for sex, age socioeconomic status and educational level).

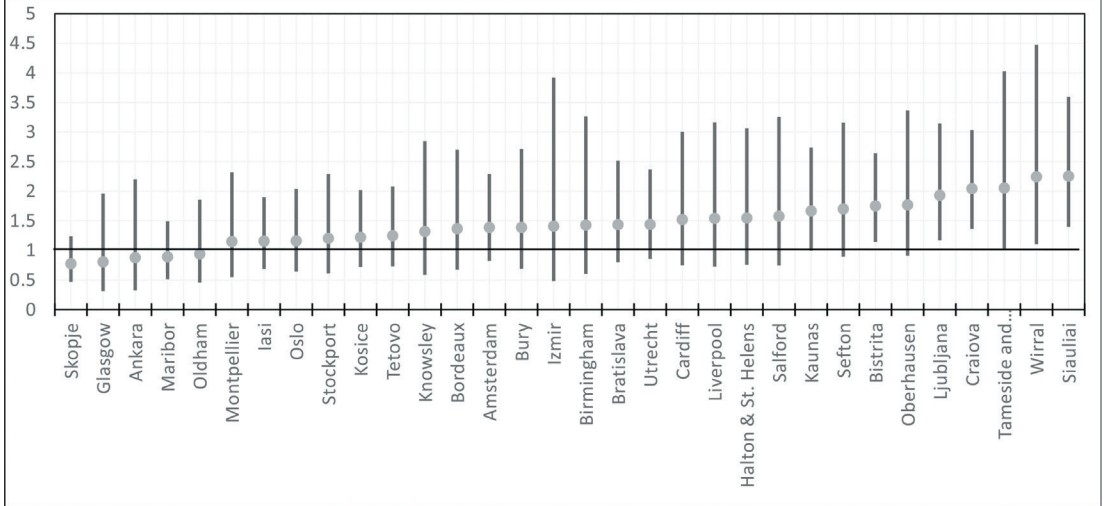

**Figure 4** Association between low back pain (LBP) and psychological distress in the EURO-URHIS 2 cities. Adjusted OR of the association between LBP and psychological distress (adjusted for sex, age socioeconomic status and educational level).

demonstrated a wide variation in LBP prevalence across European urban areas. Considering that data on LBP was collected using a previously validated question with an optimal recall period, and that the same instrument was used in all participating cities, the differences are most likely to be real. Additionally, to minimise the likelihood of information bias resulting from confusions of language or other cultural factors, all questionnaires were translated and back translated for the survey. LBP is usually managed at local general practitioners' clinics hence it is plausible that places with easier access to health services might have lower prevalence as the sufferers may receive early medical attention. This relatively high prevalence has implications for healthcare resource allocation as cities with higher proportion of their adult population suffering LBP should be expected to allocate more resources towards its care.

This research also found that the female sex, older people and people of lower socioeconomic status as well as the less educated are at greatest risk of LBP (table 1). Although similar associations with lower socioeconomic status, less education and older age have been consistent from earlier studies,[31 32] opinion is divided on the sex differences. A notable review done for the global burden of disease study found greater prevalence in males across all regions of the world.[1] However, the studies that specifically assessed 1-month period prevalence found higher proportions among females.[27–29]

Irrespective of sex, age, socioeconomic status or level of education, this study found that adult Europeans suffering from LBP have 3.5-fold higher odds of experiencing a poor state of physical health when compared with non-sufferers. This association is stronger than findings from earlier studies among European populations. A Finnish cross-sectional study reported an association between chronic pain and 'poor self-rated health' with OR ranging from 1.16 to 2.62 depending on the frequency of the pain.[33] On the other hand, a British cohort study found that background

self-rated poor health is a predictor of chronicity among LBP users of primary care.[34] Although an OR of 3.6 (95% CI: 1.9 to 6.8) was found, estimates had wide CIs as the sample size was only 180. However, whether as predictor or consequence, no previous study has compared the strength of this association across different European settings. Being a cross-sectional survey, this association, though strong, consistent across all cities and countries, and highly significant, cannot be assumed to be causal. It is also noteworthy to observe that the pattern of this association was similar to the pattern of the prevalence of LBP across countries. Implying that countries with relatively higher prevalence of LBP also tended to have stronger association with poor physical health (figure 2).

Another key finding of this research is that adult urban Europeans with LBP stand a 44% higher risk of psychological distress. An American longitudinal study similarly found that LBP at baseline was associated with a higher risk of psychological distress at the 18th month review with an OR of 1.36 (95% CI: 1.07 to 1.72).[35] The research had a much smaller sample size of 681 and was restricted to only the USA. But why is this association important? It is known that among sufferers of LBP, psychological distress predicts response to therapy and chronicity both of which have significant resource implications.[10 13] It also predicts occurrence of disability[11] and thus predicts poverty given that Europeans living with a disability are significantly poorer than those who are not.[36] This finding has potentially far reaching implications for planning public health interventions, resource allocation and social services planning and delivery. More so in countries such as the UK, and the Netherlands who despite having a comparatively low prevalence of LBP (figure 1), have significantly higher burdens of its associated poor mental health (figure 3). The same goes for urban areas such as Wirral, Craiova and Tameside and Glossop three of which have relatively low LBP prevalence but higher burdens of psychological distress associated LBP (figure 4).

Furthermore, the distribution of LBP in urban Europe varied differently from the distribution of its associated mental health burden. Implying that prevalence figures alone do not represent the associated mental health burden of LBP and likewise may not represent its other associated burdens and impact. Therefore, health policy-makers will benefit greatly from understanding the associated burdens of LBP in the context of particular urban areas to help fashion appropriate interventions among high-risk and general populations.

Globally, LBP is quite prevalent in high-income countries and in low-and-middle-income countries.[37] However, within the relatively affluent European countries, this study shows that it is the female sex, the older people, the less educated and the poorer people that are more likely to be affected by it. LBP has thus become an embodiment of engrained social mechanisms that pattern health and determinants of health in urban European societies. It has long been held that these mechanisms are not inevitable but remediable.[38] Governments at national and city levels are thus expected to make policies that will protect the vulnerable members of society from the worst effects of LBP.

Expectedly, the associated burden of LBP on physical health travels in much the same direction as its prevalence. However, its associated mental health burdens are a bit subtler. Areas of relatively low LBP prevalence may have a greater associated psychological distress. The factors that shape these differences may be connected to ease of access to care, as well as availability of support to persons and families affected by LBP. This again may be a pointer to underlying intersection of factors prevalent in those urban settings. Much as such factors deserve further exploration through research, the findings thus far should inform public health policy and action as well as the (re)structuring of social services. LBP associated with psychological distress is more likely to result in disability and eventually, poverty. Added to the demands of seeking and accessing care, families run a higher risk of being driven into further impoverishment. This potentially widens the inequalities gap in societies creating a vicious circle of disease, disability and disadvantage.

The data informing this analysis was generated from a population cross-sectional study. As a result, the temporal relationship between exposure and outcomes cannot be established by our research as both information were collected at the same time. Despite rigorous efforts to minimise bias in the EURO-URHIS 2 study, such surveys may retain the possibility of information bias on account of poor recall as well as selection bias from non-response. Additionally, our research is unable to explore if a dose–response relationship exists between LBP and psychological distress. Further research is required to answer these important questions. We also note that there was no pre-specified analysis plan in the protocol for the particular variables presented in this paper.

**Acknowledgements** The authors acknowledge all the members of the European Urban Health Indicators Survey 2 research team.

**Contributors** CIU and DP contributed to the conceptualisation and the design of this study. DP facilitated the acquisition of the data. CIU and DP contributed to the analysis and interpretation of the data, both authors contributed to the drafting and proofreading of the manuscripts and have approved of its submission. CIU holds full responsibility for the work and/or the conduct of the study.

**Funding** This work was supported by The Commonwealth Scholarship Commission (grant number NGSS-184-2015).

**Competing interests** None declared.

**Patient and public involvement** Patients and/or the public were not involved in the design, or conduct, or reporting, or dissemination plans of this research.

**Patient consent for publication** Not applicable.

**Ethics approval** Not applicable.

**Provenance and peer review** Not commissioned; externally peer reviewed.

**Data availability statement** Data are available upon reasonable request. The dataset associated with this study is available upon reasonable request by an email to: danpope@liverpool.ac.uk.

**ORCID iD**
Chukwuebuka Immanuel Ugwu http://orcid.org/0000-0001-9171-4516

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
