## [Reviewer comments · BMJ Open]

ARTICLE DETAILS

TITLE (PROVISIONAL)	Quantifying the association between psychological distress and low back pain in Urban Europe: A secondary analysis of a large cross-sectional study.
AUTHORS	Ugwu, Chukwuebuka; Pope, Daniel

VERSION 1 – REVIEW

REVIEWER	Gold, LS Pharmaceutical Outcomes Research and Policy Program, School of Pharmacy, University of Washington,
REVIEW RETURNED	16-Dec-2020

GENERAL COMMENTS	Review of “The associated mental and physical health burden of low back pain in Urban Europe: A secondary analysis of the EURO URHIS 2 Study.” General comments: This is an interesting analysis with many strengths, the most notable being the large sample size with a widespread geographic population. The variance in prevalence rates of low back pain by geographic areas is very interesting, as is the finding that psychological distress was strongly associated with prevalence of low back pain even in countries with relatively low prevalences of LBP. I have only minor issues with the methodology, described below. Abstract: If possible, it would be easier to understand the results if the referent groups were also specified where they are unclear. For example, people with primary education were at 2.28 times the odds of back pain compared to whom? Introduction: This makes a good case for the enormous burden of back pain and the benefits that this analysis brings (uniform data gathered in many different geographic areas). The sentences in the last paragraph about studying back pain among urban and rural populations were less convincing and led me to expect comparisons between urban and rural populations, which your study wasn't really designed to do. What study question were your analyses attempting to address and what hypothesis did you have about LBP and PD in urban populations among your study participants? Methods: The descriptions of the EURO-URHIS 2 data and each of the independent variables are very clear. The statistical analysis section is also very clear. I inferred from the abstract that PD and general health are the outcome variables that are predicted by the presence or absence of back pain (rather than back pain being predicted by PD and general health), but please also state this explicitly in the Methods since the cross-sectional nature of these data make that unclear. Consider not dichotomizing PD and doing linear regression with PD as a continuous outcome variable. You're losing quite a lot of information from the continuous variable by imposing a cutpoint. If a
---

	score of ≥ 4 is a standard cutpoint, please provide references. Results: Table A, supplementary, it would be easier to read if it were sorted by most to least percentages with LBP. Could you also add a column with the county of each urban area for those not familiar? The results of the prevalence of LBP stratified by age category are discussed in the text of the results but not presented in a table. I think such a table would be interesting to add if space allows, especially with the large differences between countries. For Table 1.0, please indicate in the table title that the ORs are unadjusted. It's interesting that those with primary or secondary education had higher prevalences of low back pain than those with no formal education. In the Discussion section, I would be interested in the authors' speculation about why this might be. On page 11, lines 50-52, after adjustment, only mild attenuation was noticed, not effect modification. For Figure 2.0, does the horizontal line represent an OR of 1.0? This should be labeled on the figure. Same with figure 3. Discussion: This does a good job of summarizing the study's findings and putting them in the context of other research. The authors are careful not to state any associations of cause and effect, which is appropriate for these cross-sectional data. However, a limitations paragraph is needed in this section, with the foremost limitation being that these data were collected at the same time and therefore cause and effect cannot be determined (but future studies should examine this in more detail!).
--	---

REVIEWER	Tagliaferri, Scott Deakin University
REVIEW RETURNED	09-Jan-2021

GENERAL COMMENTS	Thank-you for allowing me to review this interesting manuscript. This manuscript aimed to determine the prevalence of low back pain across urban areas in Europe. A secondary aim was to determine the association between health/distress and low back pain. I have provided some comments below which I believe will help improve the quality of the manuscript. Most of these are minor, however a couple may require a little more work to amend. I hope these ABSTRACT: Page 2 Line 42: Can you clarify which are the outcome and predictor/exposure variables? Even if the research is cross-sectional, it seems you have run a logistic regression with back pain treated as your outcome variable, while the others are exposure/predictor variables. Please amend the wording here for clarity. Page 3 Line 3: Please clarify at highest risk of which outcome. Back Pain? Page 4 Line 3: Given you have presented odds ratio the wording "highest risk" is likely not appropriate. Please amend as necessary. Page 4 Line 19: Suggest a minor change to wording to make then conclusion more succinct as widely, differently and vary all insinuate the same thing in this context. "The prevalence of low back pain, and its associations with poor physical and mental health, varies across European urban areas. SUMMARY BOX:
--

Page 5 Line 27: Can you clarify the wording here. It is not clear how you assessed the higher mental health burden? Do you mean that mental health has a higher association with low back pain in countries with a smaller prevalence of low back pain?

INTRODUCTION:

Page 6 Line 20: I'm not sure the word disappointing is appropriate. It would indicate that there is no improvement at all. Could this be termed "improvements in outcomes may be modest at best".

METHODS:

Page 8 Line 6: Were there any questions about the severity, chronicity etc of symptoms. For example, those with more severe/chronic LBP may have more severe PD etc than those without. If not then this could be a potential limitation of the study.

Page 8 Line 39: For the reader I think again, it would be worthwhile clarifying which were the exposure and outcome variables.

Page 8 Line 40: For the multivariate logistic regression, can you confirm if this included both self-rated health and PD predictor variables? If not, then this would be worth running and presenting to see if there is an effect of accounting for each predictor.

RESULTS:

Page 10 Line 16: The wording statistically significantly hard to follow. I suggest just using the word "significantly" as this implies the results were based on statistical results.

Page 11 Line 27: Please check the wording of "risk" throughout the results/manuscript given odds ratios are presented and not risk ratios.

Page 12 Line 11: For both of these tables can you please make it clearer that the 1.0 OR is the reference group in the model. Individuals may read this and assume there is not an effect of that variable. This could be done with 1.0 (Ref) or just state Ref in that box.

DISCUSSION:

Page 16 Line 23: Can you discuss what the results of the American study mean in the context of your study? Given the focus on PD as the outcome compared to you looking at back pain as the outcome. Particularly in the following sentence you state that then PD predicts back pain chronicity (outcome). Could there then be a bidirectional relationship between PD and back pain. Even though your research is cross-sectional I believe this can be discussed here for the reader.

Page 16 Line 38: Again I don't follow what you mean by a greater burden of associated mental health effects? Do you mean, areas with a low prevalence of low back pain, tend to have a greater association between PD and back pain?

Page 17 Line 31: Please add a strengths and limitations paragraph to the manuscript. For example, cross-sectional and self-report nature of LBP would be limitations.

FIGURES:

Page 24: Given you present adjusted ORs, please restate what the models were adjusted for in the figure note for ease for readers.

REVIEWER	Underwood, Martin Warwick University, Warwick Medical School
REVIEW RETURNED	15-Jan-2021

GENERAL COMMENTS	This paper adds to our knowledge on the prevalence of back pain. It does however not help our understanding of back pain and back pain disability. It is simply creating more 'why' questions. Clearly beyond scope of this paper but understanding if you ask the same question to people in Norway and Lithuania twice as many answer 'yes' in Lithuania, is really important. Is it something in the translation, the question having developed in English only, or is it a culturally driven difference in the interpretation of the question, or is there a real difference in prevalence? The authors may wish to comment on whether there may be reasons for their observations other than differences in prevalence. It might be interesting to see if global burden of disease data shows similar differences to shed some light on why this might. I have no fundamental concerns about the conduct of the study. It would , however, be nice to know if there was a pre-specified statistical analysis plan that could be included as supplementary material Whilst the overall observations on the associations between socio-economic, psychological variable, and poor self-rated health and back pain are robust, but they are not novel findings. I would be inclined to give them less prominence for this reason. The differences in strength of these associations by nation is novel and of interest – this could be explored more in the discussion. More could be said about how these differences might be explain differences in overall prevalence – why should there be no association between psychological distress in Norway and an odds ratio of around two for this in Lithuania? More up to date references are available for prevalence of, and disability caused by, low back pain. Notably, reference 36 (Volinn) is out of date. It is now generally believed prevalence of low back pain is high in both developed and less developed nations. See for example a fairly recent systematic review of back pain prevalence studies from Africa https://pubmed.ncbi.nlm.nih.gov/30037323/. I would prefer less use of non-standard abbreviations. Specifically, PD for psychological distress. I kept needing to remind myself I was not reading a Parkinson's disease study. Also why say UA instead of urban area. I am not convinced that the approach of reporting individual country prevalences when compared to the European average is robust. I may be misunderstanding, but presumably the European data will include the all the individual country data. More detail on this analysis is needed and revie by a statistical expert. Although, I think the data stand as they are without any need for any statistical gloss. Whilst it is possible to work out what the figures mean they could be made easier to understand with properly labelled Y axes and actual values attached to bars in all four figures.
--

VERSION 1 – AUTHOR RESPONSE

Section	Review Comment	Authors' feedback
Reviewer 1: Dr LS Gold – University of Washington	Review of “The associated mental and physical health burden of low back pain in Urban Europe: A secondary analysis of the EURO URHIS 2 Study.” General comments: This is an interesting analysis with many strengths, the most notable being the large sample size with a widespread geographic population. The variance in prevalence rates of low back pain by geographic areas is very interesting, as is the finding that psychological distress was strongly associated with prevalence of low back pain even in countries with relatively low prevalences of LBP. I have only minor issues with the methodology, described below.	Comments noted and appreciated.
Abstract	If possible, it would be easier to understand the results if the referent groups were also specified where they are unclear. For example, people with primary education were at 2.28 times the odds of back pain compared to whom?	These comments are noted. Given the strict word count for the abstract section, it may not be feasible to provide further explanation of the comparator categories in the abstract. The sentence has been rephrased to read: The overall European Prevalence of low back pain was 44.6% (43.9-45.3) widely ranging from 33.4% in Norway to 67.7% in Lithuania. After accounting for sex, age, socio-economic status, and formal education, adults in urban Europe suffering low back pain had higher odds of psychological distress aOR 1.44 (1.32-1.58) and poor self-rated health aOR 3.54 (3.31-3.80). These associations varied widely between participating countries and cities. See page 2 lines 19 – 20, page 3 lines 1 - 3

Introduction:	This makes a good case for the enormous burden of back pain and the benefits that this analysis brings (uniform data gathered in many different geographic areas). The sentences in the last paragraph about studying back pain among urban and rural populations were less convincing and led me to expect comparisons between urban and rural populations, which your study wasn't really designed to do.	This has been modified. The sentences in the last paragraph of the introduction now read thus: In addition, although LBP affects all population settings, the burden is expected to differ in urban settings. See page 5 lines 6 – 7
	What study question were your analyses attempting to address and what hypothesis did you have about LBP and PD in urban populations among your study participants?	The study question has been included in the introduction section and reads as follows: Amongst adults dwelling in urban Europe, what is the risk of psychological distress in those with low back pain compared to those without low back pain. See page 5 lines 11 - 12 Further response: Population: adults dwelling in urban Europe Exposure: low back pain Comparator: those without low back pain Outcome: psychological distress Null Hypothesis: there is no significant difference in the risk of psychological distress among adults in urban Europe who suffer low back pain compared to

		those who do not.
Methods:	The descriptions of the EURO-URHIS 2 data and each of the independent variables are very clear. The statistical analysis section is also very clear. I inferred from the abstract that PD and general health are the outcome variables that are predicted by the presence or absence of back pain (rather than back pain being predicted by PD and general health), but please also state this explicitly in the Methods since the cross-sectional nature of these data make that unclear.	Comments noted. The exposure and outcome variables have been stated explicitly in the methods section. Page 6 line 14 - 18 and reads thus: The exposure (predictor) variable for this study was low back pain whilst psychological distress and self-rated health were the main outcome variables. Data was also collected on potential socio-demographic confounders known to have a relationship with the predictor. These include: age, sex, level of formal education attained, and socio-economic status. All these were accounted for through multivariable regression.
	Consider not dichotomizing PD and doing linear regression with PD as a continuous outcome variable. You're losing quite a lot of information from the continuous variable by imposing a cutpoint. If a score of ≥ 4 is a standard cutpoint, please provide references.	The relevant reference has been provided. The Swedish validation study cited aimed to validate the GHQ-12 for the assessment of mental health at population level. They found that the best cut-off point with excellent discriminant validity was ≥ 4. https://www.tandfonline.com/doi/abs/10.1080/08039488.2016.1246608
Results:	Table A, supplementary, it would be easier to read if it were sorted by most to least percentages with LBP. Could you also add a column with the county of each urban area for those not familiar?	The table has been sorted by ascending order of magnitude and the column for country added. See Supplementary materials Tables A,

		B, and C.
	The results of the prevalence of LBP stratified by age category are discussed in the text of the results but not presented in a table. I think such a table would be interesting to add if space allows, especially with the large differences between countries.	The table has not been added for space. Required table has been included in the supplementary materials as table c and cited in the text in page 10 line 1.
	For Table 1.0, please indicate in the table title that the ORs are unadjusted	This has been modified as recommended. The title of the table 1.0 no reads: 'Unadjusted Odds ratios of LBP associated with socio-demographic variables' See page 11 table 1.0
	It's interesting that those with primary or secondary education had higher prevalences of low back pain than those with no formal education. In the Discussion section, I would be interested in the authors' speculation about why this might be.	Comments noted. The authors have not given prominence to discussions around the socio-demographic factors associated with LBP. They were identified as potential confounders that had to be accounted for through multivariable regression. Hence speculations about the mechanisms of the interesting association of LBP and educational status are not pursued.
	On page 11, lines 50-52, after adjustment, only mild attenuation was noticed, not effect modification.	The relevant section has been reworded as recommended. See page 11 lines 6 - 7
	For Figure 2.0, does the horizontal line represent an OR of 1.0? This should be labeled on the figure. Same with figure 3.	The figures 2.0 and 3.0 have been appropriately labelled (y axes calibrated).
Discussion:	This does a good job of summarizing the study's findings and putting them in the context of other research. The authors are careful not to state any associations of cause and effect, which is appropriate	Comments noted. We have included a section on limitations of our study on page 17 lines

	for these cross-sectional data. However, a limitations paragraph is needed in this section, with the foremost limitation being that these data were collected at the same time and therefore cause and effect cannot be determined (but future studies should examine this in more detail!).	4 - 10 and it reads thus: The data informing this analysis was generated from a population cross-sectional study. As a result, the temporal relationship between exposure and outcomes cannot be established by our research as both information were collected at the same time. Despite rigorous efforts to minimize bias in the EURO URHIS 2 study, such surveys may retain the possibility of information bias on account of recall as well as selection bias from non-response. Additionally, our research is unable to explore if a dose-response relationship exists between LBP and psychological distress. Further research is required to answer these questions.
Reviewer 2: Dr Scott Tagliaferri – Deakin University	Comments to the Author: Thank-you for allowing me to review this interesting manuscript. This manuscript aimed to determine the prevalence of low back pain across urban areas in Europe. A secondary aim was to determine the association between health/distress and low back pain. I have provided some comments below which I believe will help improve the quality of the manuscript. Most of these are minor, however a couple may require a little more work to amend. I hope these	Comments noted and appreciated.
ABSTRACT:	Page 2 Line 42: Can you clarify which are the outcome and predictor/exposure variables? Even if the research is cross-sectional, it seems you have run a logistic regression with back pain treated as your outcome variable, while the others are exposure/predictor variables. Please amend the wording here for clarity.	The exposure/predictor variable in our research is low back pain and the outcomes are psychological distress and self-rated health. The analysis in which we considered LBP as the outcome (table 1.0) was to identify potential socio-demographic confounders which would then be adjusted for in the Multivariable regression (table 2.0).
	Page 3 Line 3: Please clarify at highest risk of which outcome. Back Pain?	Sentence has been modified as recommended to specify 'low back pain'.
	Page 4 Line 3: Given you have presented odds ratio the wording "highest risk" is	Expression highest risk modified as

	likely not appropriate. Please amend as necessary.	recommended no reads: 'high odds'
	Page 4 Line 19: Suggest a minor change to wording to make then conclusion more succinct as widely, differently and vary all insinuate the same thing in this context. "The prevalence of low back pain, and its associations with poor physical and mental health, varies across European urban areas.	Comments noted. Modifications have been effected. The aim was to communicate a variation in the prevalence of LBP as well as a difference in the variation pattern of its associated psychological distress. The recommended phrasing has been adopted. See page 3 lines 5 - 6
SUMMARY BOX:	Page 5 Line 27: Can you clarify the wording here. It is not clear how you assessed the higher mental health burden? Do you mean that mental health has a higher association with low back pain in countries with a smaller prevalence of low back pain?	Summary box has been removed based on recommendations from the journal editorial unit.
INTRODUCTION:	Page 6 Line 20: I'm not sure the word disappointing is appropriate. It would indicate that there is no improvement at all. Could this be termed "improvements in outcomes may be modest at best".	Sentence has been modified as recommended. Now reads: "improvements in outcomes may be modest at best". See page 4 line 16
METHODS:	Page 8 Line 6: Were there any questions about the severity, chronicity etc of symptoms. For example, those with more severe/chronic LBP may have more severe PD etc than those without. If not then this could be a potential limitation of the study.	This has been acknowledged as a limitation of our research. See page 17 lines 4 - 10
	Page 8 Line 39: For the reader I think again, it would be worthwhile clarifying which were the exposure and outcome variables.	This has been clarified Page 6 lines 14 - 18 and reads thus: The key exposure (predictor) variable for this study was low back pain whilst psychological distress and self-rated

		health were the main outcome variables.
	Page 8 Line 40: For the multivariate logistic regression, can you confirm if this included both self-rated health and PD predictor variables? Of not, then this would be worth running and presenting to see if there is an effect of accounting for each predictor.	Comments noted. We have treated self-rated health and psychological distress as outcome variables and LBP as the predictor/exposure based on existing literature and the theory of biological plausibility.
RESULTS:	Page 10 Line 16: The wording statistically significantly hard to follow. I was suggest just using the word "significantly" as this implies the results was based off statistical results.	The wording has been amended as recommended.
	Page 11 Line 27: Please check the wording of "risk" throughout the results/manuscript given odds ratio are presented and not risk ratios.	Wordings have been modified as recommended now read 'higher odds'
	Page 12 Line 11: For both of these tables can you please make it clearer that the 1.0 OR is the reference group in the model. Individuals may read this and assume there is not an effect of that variable. This could be done with 1.0 (Ref) or just state Ref in that box.	Thank you. 'Ref' has been included in parenthesis for the reference groups. See page 11 table 1.0, page 12 table 2.0
DISCUSSION:	Page 16 Line 23: Can you discuss what the results of the American study mean in the context of your study? Given the looks at PD as the outcome compared to you looking at back pain as the outcome. Particularly in the following sentence you state that then PD predicts back pain chronicity (outcome). Could there then be a bidirectional relationship between PD and back pain. Even though your research is cross-sectional I believe this can be discussed here for the reader.	Comments noted. LBP is predictor whilst PD is outcome in our study.

	Page 16 Line 38: Again I don't follow what you mean by a greater burden of associated mental health effects? Do you mean, areas with a low prevalence of low back pain, tend to have a greater association between PD and back pain?	The phrasing has been modified as recommended. Now reads: 'areas with a low prevalence of low back pain, tend to have a greater associated Psychological distress' See page 16 lines 15 - 17
	Page 17 Line 31: Please add a strengths and limitations paragraph to the manuscript. For example, cross-sectional and self-report nature of LBP would be limitations.	A section on limitations has been added. See page 17 lines 4 - 10
FIGURES:	Page 24: Given you present adjusted ORs, please restate what the models were adjusted for in the figure note for ease for readers.	The variables for which the models adjusted have been included. See figures 2.0, 3.0, and 4.0

Reviewer 3: Prof martin Underwood – Warwick University	Comments to the Author: This paper adds to our knowledge on the prevalence of back pain. It does however not help our understanding of back pain and back pain disability. It is simply creating more 'why' questions. Clearly beyond scope of this paper but understanding if you ask the same question to people in Norway and Lithuania twice as many answer 'yes' in Lithuania, is really important. Is it something in the translation, the question having developed in English only, or is it a culturally driven difference in the interpretation of the question, or is there a real difference in prevalence? The authors may wish to comment on whether there may be reasons for their observations other than differences in prevalence. It might be interesting to see if global burden of disease data shows similar differences to shed some light on why this might.	The comments are noted. On the issue of the possibility of information bias on account of the nuances of translation, we have the following comments to offer. It was required that questions be selected from existing standardized or validated instruments used for health and lifestyle surveys in local populations across Europe. Ideally it was required that questions be selected from pre-existing surveys that have:  1. Been administered across more than one country. 2. Focused on adults aged 18 years and over. 3. Used short, validated questions. 4. Preferably used questions that have been translated into European languages. 5. Used a self-administered postal questionnaire approach (in the absence of suitable examples being identified from this source surveys using face-to-face or telephone interview approaches would be selected). Five working groups were established to identify as many validated sources as possible for Urban Health Indicators relevant to each of the 5 domains. The working groups consisted of steering group members from Euro-Urhis 2 who either had previous experience of these areas of research or who expressed an interest in working within this domain. Leads for each working group were identified who had experience of survey-based methodology.
---	--	---

		Validated sources for UHIs used by the working groups to identify relevant questions included: (1) European Health Interview Survey (EHIS) Questionnaire. The questionnaire was adopted on 22 November 2006 by the Eurostat Working Group on Public Health Statistics. It is expected to be administered every 5 years from 2007/2008 onwards: It is composed of 4 modules:  - EHSM (health status) - EHDM (health determinants) = Lifestyle - EHCM (health care) - EBM (background variables) It was tested and validated by Institute of Public Health, Brussels during 2005-2008. It has been translated into Danish, English, French, German, Hungarian and Italian. Questions were delivered in an interview format. Reports of varying response rates were noted during the implementation of EHIS methodology in different countries (ranging from 5% in Brussels to 90% in Finland), with a clear North to South gradient being displayed. (2) FINBALT Health Monitor is a collaborative system for monitoring health related behaviour, practices and lifestyles in Estonia, Finland, Latvia, and Lithuania. It monitors behaviour such as smoking, alcohol consumption, food habits and physical activity. The core questionnaire - common to all participating countries - has about 100 questions. It has been running since 1978 in Finland with other countries joining in the 1990s. Age range targeted 15-64-years, with a separate
--	--	---

		questionnaire for the older population 65- 84. The survey was carried out simultaneously in all Baltic countries and Finland for the first time in spring 1998. The Finbalt questionnaire has been used to develop health monitoring survey for WHO CINDI (Countrywide Integrated Non-communicable Disease Intervention) programme. It is administered centrally by KTL Finlands National Institute for Health and Welfare. The most recent available edition is dated 2004. (3) HEPRO (Health and social well-being in the Baltic Sea Region) The project was run from 2005-2008, partners from 8 countries, Norway, Sweden, Denmark, Finland, Estonia, Latvia, Lithuania and Poland participated in the project The HEPRO survey model, a questionnaire was answered by 33000 respondents in the Baltic Sea Region during October-November 2006. (Finland and Sweden did not take part in the final survey) Age range targeted 16-79 years. The HEPRO project focussed on developing a questionnaire that included core indicators relevant from a modern concept of health and health promotion, combining the biomedical, epidemiological and psychosocial descriptions of health. As well as an English master copy the questions were variously in different languages: Estonian, Latvian, Lithuanian, Polish, Russian, Danish, Swedish and Norwegian, (4) NWPHO (North West Public Health Observatory) Core Questions and Methods. The NWPHO was asked to provide a framework for the conduct of lifestyle surveys across the North West region of England. The questions were further tested against European targets. Following consultation around the region a set of questions has been
--	--	--

		developed that is recommended as a minimum core that will help to collect data on key lifestyle topics. The questions were chosen for being able to provide data to support target setting and monitoring and if they had been previously validated. A final report was written in 2007 and subsequently the questions were used to conduct a regional lifestyle survey. The age range targeted 16 years and older. Whilst there has been no translation, options for translation have been offered. At a 3-day steering group meeting the lead from each working group presented a paper detailing the items from their domain that had been deemed appropriate for measuring UHIs within this domain. Through discussion, items identified as being the most appropriate, practical and generic (in terms of being understood in multiple European countries after translation) were identified for inclusion in the Euro-Urhis data collection tool. If a previously validated source could not be found for any UHI necessary for Euro-Urhis 2 suitable questions were drafted by core members of the steering group and management committee. These were then circulated to members of the steering group and to city partners for commentary. Further details of translation and back-translation of the survey instruments are published on the Manchester University website: http://www.urhis.eu/scopeofwork/wp4-protocols/
	I have no fundamental concerns about the conduct of the study. It would, however, be nice to know if there was a pre-specified statistical analysis plan that could be included as supplementary	No pre-specified statistical analysis plan was included in the EURO URHIS 2 survey protocol as regards our variables of interest.

	material	
	Whilst the overall observations on the associations between socio-economic, psychological variable, and poor self-rated health and back pain are robust, but they are not novel findings. I would be inclined to give them less prominence for this reason.	Comments have been noted. The section in the abstract and discussion that seemed to give prominence to these have been modified. See page 2 lines 19 – 20, page 3 lines 1 - 3
	The differences in strength of these associations by nation is novel and of interest – this could be explored more in the discussion. More could be said about how these differences might be explain differences in overall prevalence – why should there be no association between psychological distress in Norway and an odds ratio of around two for this in Lithuania?	Further discussion has been added to this section. See Page 16 lines 17 to 20.
	More up to date references are available for prevalence of, and disability caused by, low back pain. Notably, reference 36 (Volinn) is out of date. It is now generally believed prevalence of low back pain is high in both developed and less developed nations. See for example a fairly recent systematic review of back pain prevalence studies from Africa https://pubmed.ncbi.nlm.nih.gov/30037323/ .	Comments noted and appreciated. New reference suggestion is appreciated and has been included in place of the outdated reference. The corresponding section in the discussion has also been rephrased to reflect the current evidence. Page 16 Line 7 now reads: Globally, low back pain is quite prevalent in high and low-and-middle-income countries.
	I would prefer less use of non-standard abbreviations. Specifically, PD for psychological distress. I kept needing to remind myself I was not reading a Parkinson's disease study. Also why say UA instead of urban area.	Comments noted. UA and PD abbreviations have now all been written out in full to read Urban areas and psychological distress.
	I am not convinced that the approach of reporting individual country prevalences	The comments are noted.

	when compared to the European average is robust. I may be misunderstanding, but presumably the European data will include the all the individual country data. More detail on this analysis is needed and review by a statistical expert. Although, I think the data stand as they are without any need for any statistical gloss.	We now refer to the 'overall prevalence' rather than 'European average'.
	Whilst it is possible to work out what the figures mean they could be made easier to understand with properly labelled Y axes and actual values attached to bars in all four figures.	All the Y-axes of the figures have been labelled and values added to the bars of figure 1.0. for figures 2.0 and 3.0, addition of values to the lines was not suitable thus a data table has been added to the figure to show the actual aORs and the 95% CIs.

VERSION 2 – REVIEW

REVIEWER	Gold, LS Pharmaceutical Outcomes Research and Policy Program, School of Pharmacy, University of Washington,
REVIEW RETURNED	10-May-2021

GENERAL COMMENTS	Thanks, the authors have done a commendable job responding to my comments.
--

REVIEWER	Tagliaferri, Scott Deakin University
REVIEW RETURNED	18-May-2021

GENERAL COMMENTS	I would like to thank the authors for taking the time to respond to my comments in detail. I have now reviewed these and they have been adequately implemented. How the exposures and outcomes were treated in the statistical models is now clearer. Outside of the responses, I have some minor comments that should be addressed. I apologise if I missed any of these in the original review. RESULTS: Page 10 Line 1: Throughout the results please present the 95% confidence intervals anytime a value from statistical models is reported to allow readers to easily determine the precision of the estimate. EG on line 19 "Whereas those with No formal education had a 44% increase in risk." The 95% CI should be presented with this. Page 10 Line 10: The wording needs to be updated from 'risk' to 'odds' given the odds ratios were used for consistency. This also needs to be updated in subsequent paragraphs. DISCUSSION:
---

	Page 16 Line 7: Low back pain can be abbreviated here. FIGURES: Figure 2 & 3: I found the high, low, close terms confusing. Would there be any issue with using something like UCI (upper confidence interval), LCI (lower confidence interval), and aOR (adjusted odds ratio) to be related to the terminology used throughout the manuscript. I also believe the aOR should be presented on the top line not the bottom. Figure 4: There is a figure caption but the terms don't seem to be presented in the figure. Please add these to the figure, or remove the terms from the caption. If the terms are removed from the caption "Adjusted Odds ratio and 95% confidence intervals of the Association between Low back pain and Psychological distress (adjusted for sex, age socioeconomic status, and educational level)" should remain to explain the figure.
--	--

REVIEWER	Underwood, Martin Warwick University, Warwick Medical School
REVIEW RETURNED	07-May-2021

GENERAL COMMENTS	At initial submission I made the following comment 'Clearly beyond scope of this paper but understanding if you ask the same question to people in Norway and Lithuania twice as many answer 'yes' in Lithuania, is really important. Is it something in the translation, the question having developed in English only, or is it a culturally driven difference in the interpretation of the question, or is there a real difference in prevalence? The authors may wish to comment on whether there may be reasons for their observations other than differences in prevalence. It might be interesting to see if global burden of disease data shows similar differences to shed some light on why this might be.' The authors have provided a very long description of the selection of measures for the study in response to referees. But they have not sought to address this point in the text. Some description of the lengths they went to address possibility of information bias and how confident they are that the differences observed are true differences. I think in the limitations it should be stated that there was not a pre-specified statistical analysis plan The updated figures do not appear to have been included in the re-submission
---

VERSION 2 – AUTHOR RESPONSE

Reviewer 2	
I would like to thank the authors for taking the time to respond to my comments in detail. I have now reviewed these and they have been adequately implemented. How the exposures and	Noted and appreciated.

outcomes were treated in the statistical models is now clearer. Outside of the responses, I have some minor comments that should be addressed. I apologize if I missed any of these in the original review.	
Page 10 Line 1: Throughout the results, please present the 95% confidence intervals anytime a value from statistical models is reported to allow readers to easily determine the precision of the estimate. EG on line 19 "Whereas those with No formal education had a 44% increase in risk." The 95% CI should be presented with this	Thank you for these comments. All the data from models have now been presented with their associated 95% Confidence intervals.
Page 10 Line 10: The wording needs to be updated from 'risk' to 'odds' given the odds ratios were used for consistency. This also needs to be updated in subsequent paragraphs	The word 'risk' has now been replaced with 'odds'
Page 16 Line 7: Low back pain can be abbreviated here.	'Low back pain' has been abbreviated to 'LBP'
Figure 2 & 3: I found the high, low, close terms confusing. Would there be any issue with using something like UCI (upper confidence interval), LCI (lower confidence interval), and aOR (adjusted odds ratio) to be related to the terminology used throughout the manuscript. I also believe the aOR should be presented on the top line not the bottom.	Apologies for this. Our graphs were created in MS Excel and we are not able to alter the formatting and labelling of adjoining tables of this particular type of chart (high-low-close).
Figure 4: There is a figure caption but the terms don't seem to be presented in the figure. Please add these to the figure, or remove the terms from the caption. If the terms are removed from the caption "Adjusted Odds ratio and 95% confidence intervals of the Association between Low back pain and Psychological distress (adjusted for sex, age socioeconomic status, and educational level)" should remain to explain the figure.	Thank you. The terms that did not appear in the chart have been removed.
Reviewer: 3 At initial submission I made the following comment 'Clearly beyond scope of this paper but understanding if you ask the same question to	Thank you for these comments. An explanation on some of the efforts to prevent such confusion has now been added to the text discussion section on page 14 Lines 6 – 8. It reads thus:

people in Norway and Lithuania twice as many answer 'yes' in Lithuania, is really important. Is it something in the translation, the question having developed in English only, or is it a culturally driven difference in the interpretation of the question, or is there a real difference in prevalence? The authors may wish to comment on whether there may be reasons for their observations other than differences in prevalence. It might be interesting to see if global burden of disease data shows similar differences to shed some light on why this might be.' The authors have provided a very long description of the selection of measures for the study in response to referees. But they have not sought to address this point in the text. Some description of the lengths they went to address possibility of information bias and how confident they are that the differences observed are true differences.	Additionally, to minimize the likelihood of information bias resulting from confusions of language or other cultural factors, all questionnaires were translated and back-translated for the survey.
I think in the limitations it should be stated that there was not a pre-specified statistical analysis plan	This has been included in the limitations section. Page 17 lines 15 - 16
The updated figures do not appear to have been included in the re-submission	The updated figures have been uploaded as separated files as per journal requirement. Not in the main text.

VERSION 3 – REVIEW

REVIEWER	Tagliaferri, Scott Deakin University
REVIEW RETURNED	05-Sep-2021
GENERAL COMMENTS	I would like to thank the authors for adequately addressing my last round of comments and I have nothing further to add.